# Finite Element Analysis of Nanoindentation Responses in Bi₂Se₃ Thin Films

**Shu-Wei Cheng [1,†], Bo-Syun Chen [2,†], Sheng-Rui Jian [1,3,4,*], Yu-Min Hu [3,*], Phuoc Huu Le [5,*], Le Thi Cam Tuyen [6], Jyh-Wei Lee [7] and Jenh-Yih Juang [8]**

[1] Department of Materials Science and Engineering, I-Shou University, Kaohsiung 84001, Taiwan

[2] Department of Mechanical and Electro-Mechanical Engineering, National Sun Yat-sen University, Kaohsiung 80424, Taiwan

[3] Department of Applied Physics, National University of Kaohsiung, Kaohsiung 81148, Taiwan

[4] Department of Fragrance and Cosmetic Science, College of Pharmacy, Kaohsiung Medical University, Kaohsiung 80708, Taiwan

[5] Department of Physics and Biophysics, Faculty of Basic Sciences, Can Tho University of Medicine and Pharmacy, Can Tho City 94000, Vietnam

[6] Department of Chemical Engineering, College of Engineering Technology, Can Tho University, Can Tho City 900000, Vietnam

[7] Department of Materials Engineering, Ming Chi University of Technology, New Taipei City 24301, Taiwan

[8] Department of Electrophysics, National Yang Ming Chiao Tung University, Hsinchu 30010, Taiwan

\* Correspondence: srjian@gmail.com (S.-R.J.); ymhu@nuk.edu.tw (Y.-M.H.); lhuuphuoc@ctump.edu.vn (P.H.L.)

† These authors contributed equally to this work.

**Abstract:** In this study, the nanoindentation responses of Bi₂Se₃ thin film were quantitatively analyzed and simulated by using the finite element method (FEM). The hardness and Young's modulus of Bi₂Se₃ thin films were experimentally determined using the continuous contact stiffness measurements option built into a Berkovich nanoindenter. Concurrently, FEM was conducted to establish a model describing the contact mechanics at the film/substrate interface, which was then used to reproduce the nanoindentation load-depth and hardness-depth curves. As such, the appropriate material parameters were obtained by correlating the FEM results with the corresponding experimental load-displacement curves. Moreover, the detailed nanoindentation-induced stress distribution in the vicinity around the interface of Bi₂Se₃ thin film and *c*-plane sapphires was mapped by FEM simulation for three different indenters, namely, the Berkovich, spherical and flat punch indenters. The results indicated that the nanoindentation-induced stress distribution at the film/substrate interface is indeed strongly dependent on the indenter's geometric shape.

**Keywords:** Bi₂Se₃ thin film; nanoindentation; finite element method

## 1. Introduction

With its unique quintuple layer structure, Bi₂Se₃ behaves like a narrow band gap semiconductor with excellent thermoelectric properties near room temperature, as well as a 3D topological insulator with a large bulk band gap (0.3 eV) and topologically protected surface state [1,2]. Such rich emergent physical properties automatically invite tremendous research interest due to its potential applications in a wide range of next generation devices [3,4]. However, while most of the research has been focusing on the thermoelectric [5] and transport properties [6], research on the mechanical characterizations has not drawn equal attention. Since for most device fabrication processes, contact-induced damage may significantly affect the properties of the films upon which devices are made, which in turn would substantially influence the performance of the devices, a comprehensive understanding of the mechanical properties of Bi₂Se₃, especially how the film reacts when under localized compressive stress, is indispensable for fabricating efficient and endurable devices.

Nanoindentation is a popular technique being widely adopted to obtain prominent mechanical property parameters, such as the Young's modulus and hardness, with various materials, especially for film/substrate systems [7–10]. However, nanoindentation itself does not provide information about the mechanism of nanoindentation-induced deformation mechanisms and stress distribution within the film/substrate system in a direct manner, which, from a practical point of view, is more relevant to stress-induced deteriorations during device processing. In this respect, finite element modeling (FEM) might serve in a complementary role not only in revealing the nanomechanical properties of thin films [11,12], but also in unveiling the stress distribution within the film and at the interface during nanoindentation [13,14], or even in explaining the crack formation and delamination phenomenon [15,16]. Nevertheless, simulating the nanoindentation process is often a highly complicated task due to the nonlinear behavior of the process. For instance, Lichinchi et al. [17] reported the results of combining FEM simulations with nanoindentation using a Berkovich indenter tip on TiN thin film deposited on high-speed steel and concluded that no apparent differences are observed between the experimental load-displacement (*P-h*) curves and those obtained from FEM using the 2D axisymmetric model with a conical indenter and/or the 3D pyramidal model. The results suggest the feasibility of combining FEM simulation and actual indentation measurements to extract the prominent parameters for a more detailed understanding of the contact-induced mechanistic behaviors.

In this study, a combination of experiments and 2D axisymmetric FEM analysis on the nanoindentation responses of $Bi_2Se_3$ thin films deposited on c-plane sapphires is investigated. The nanomechanical properties, e.g., hardness, Young's modulus, as well as the *P-h* curves, are experimentally measured. The FEM analysis is carried out to simulate the experimentally measured P-h and hardness-depth curves. Moreover, the effects of the indenters' geometries, for Berkovich, flat punch and spherical indenters, on the interfacial stress distribution of the $Bi_2Se_3$ thin film/c-plane sapphire system during the nanoindentation processes are also discussed. The fact that the present FEM simulations are able to replicate the main features of actual nanoindentation experiments evidently validates the feasibility of reliably developing the mechanical deformation of $Bi_2Se_3$ thin films deposited on *c*-plane sapphire substrate using FEM simulation, which has been largely missing in previous investigations. Moreover, the comparisons performed on indenters with various geometries may also provide an efficient means to evaluate the contact-induced deformation encountered in practical device fabrication processes, wherein the shape of the contact tip is often case dependent.

## 2. Materials and Methods

The $Bi_2Se_3$ thin films investigated in this work were grown on *c*-plane sapphire substrates by using pulsed laser deposition method with an average thickness of about 360 nm. The details of growth procedures in preparing these $Bi_2Se_3$ thin films can be found elsewhere [7].

The nanoindentation measurements were carried out on a Nanoindenter MTS NanoXP® system (MTS Cooperation, Nano Instruments Innovation Center, Oak Ridge, TN, USA) with a diamond pyramid-shaped Berkovich-type indenter tip, whose radius of curvature is 50 nm. The measurements were performed with a continuous stiffness mode [18] and a constant nominal strain rate of $0.05 \text{ s}^{-1}$. The hardness (*H*) and Young's modulus (*E*) of $Bi_2Se_3$ thin films, calculated from the P-h curves based on the analytic method developed by Oliver and Pharr [19], are about 1.8 GPa and 70 GPa, respectively.

To investigate the nanoindentation-induced deformation behaviors of $Bi_2Se_3$ thin films on the c-plane sapphire substrate, the film–substrate structure was modelled by FEM. In particular, not only the *P-h* curves, but also the distribution of stress fields, strain fields, and the profile of the indentation are also analyzed using FEM under nanoindentation. In this model, the diameter and thickness of the sapphire substrate are both assumed to be 10,000 nm and the thickness of a $Bi_2Se_3$ thin film is taken as 360 nm. FEM simulations

are made with the 2D axisymmetric conical indenter, which is equivalent to the Berkovich indenter, and the indentation curves are evaluated. The actual indenter was constructed from a diamond with a height of 2200 nm. To define an axis symmetrical model, an equivalent conical indenter with a semi apical angle of 70.3° having the same contact area as the Berkovich indenter was used [20]. In addition, simulations with flat punch and spherical indenters are also included for comparison, as depicted schematically in Figure 1. The geometrical dimensions of the flat punch indenter tip are: 1200 nm in height and 1600 nm in diameter [21]. For the spherical indenter, the radius of the tip is 1500 nm [22].

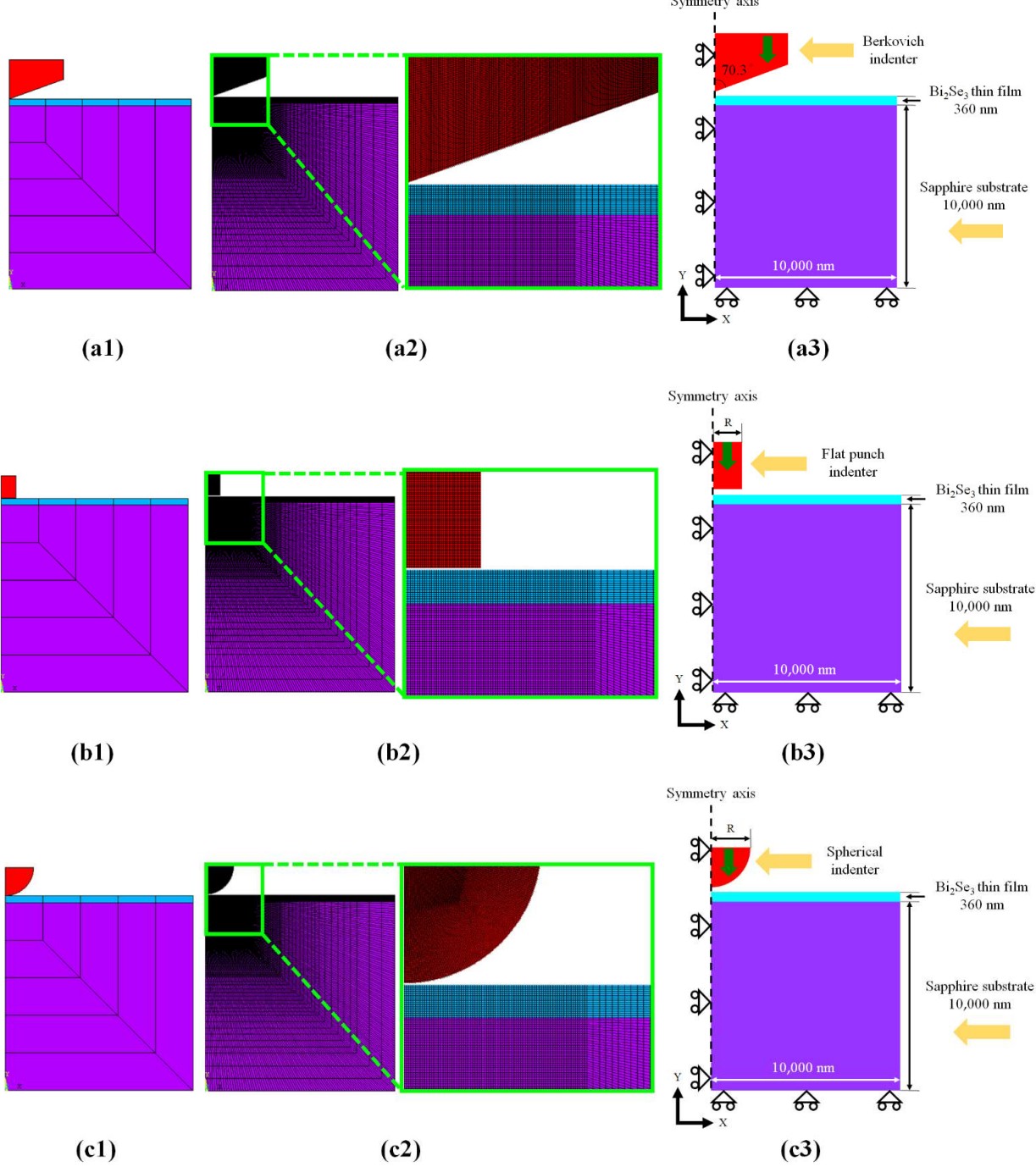

**Figure 1.** The entire specimen with (**a1**) Berkovich, (**b1**) flat punch and (**c1**) spherical indenters for the 2D model before meshing. Mesh of the entire specimen with (**a2**) Berkovich, (**b2**) flat punch and (**c2**) spherical indenters for the 2D model. The geometry, boundary, and load conditions of (**a3**) Berkovich, (**b3**) flat punch and (**c3**) spherical indenters.

By using an applied force of 0.12 mN, the maximum depth can be obtained for the analysis of mesh convergence. The displacement is measured along the Y-axis in Figure 2. The result is the displacement of the center of the sample, which is used to verify the maximum depth of the experiment. When the boundary conditions apply the same load, the increase of the mesh does not affect the maximum displacement, which means that the number of meshes in the model reaches a convergent state. A high-density mesh is performed in this critical area to confirm the accuracy of the simulation analysis. From the maximum curve, it is known that the number of elements is more than 20,000 after reaching the convergence state, as shown in Figure 2. We increase the mesh density near the contact point of the indenter tip and the film to observe the changes in the vicinity of this point, as shown in Figure 1(a2,b2,c2). The model uses a mapped mesh, not a free mesh, in order to be able to control the number and quality of elements. Material properties of $Bi_2Se_3$ used in the model were obtained from the experimental data. The simplest relationship between hardness ($H$) and yield stress ($\sigma_y$) is approximated as $H \approx 3\sigma_y$ according to Gupta et al. [14] and is used as the presumed value. Material properties of the indenter and substrate required for modeling are listed in Table 1. It is noted that the bi-linear model requires the Young's modulus to represent the elastic phase and the tangent modulus to describe the plastic phase. Nevertheless, from the nanoindentation experiments, we can only obtain the Young's modulus and hardness, not the tangent modulus, for the simulation input. Because of the lack of the tangent modulus, our simulations became sort of elastic perfect-plastic, although our material model was not only considering the linear flexible part. This is believed to be the primary reason why the obtained stress contours were not homogeneous (see below).

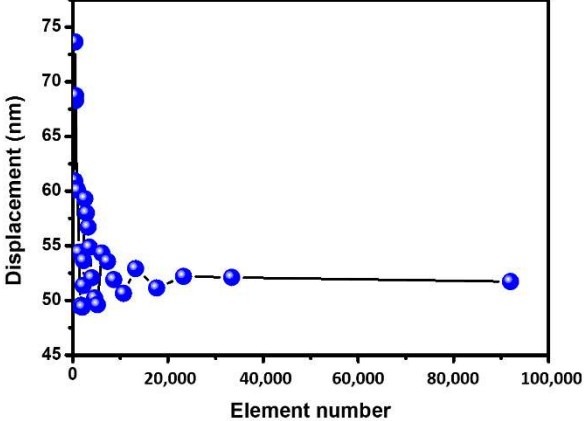

**Figure 2.** The element number convergence analysis, indicating the mesh independence above element number of 20,000.

**Table 1.** Mechanical properties of diamond indenter, $Bi_2Se_3$ film and sapphire substrate.

| Component | Materials | Properties |
|---|---|---|
| Indenter | Diamond [19] | Elastic modulus: 1140 GPa<br>Poisson's ratio: 0.07 |
| Thin film | $Bi_2Se_3$ [this work] | Elastic modulus: 70 GPa<br>Poisson's ratio: 0.25<br>Initial yield stress: 0.6 GPa |
| Substrate | Sapphire [23] | Elastic modulus: 450 GPa<br>Poisson's ratio: 0.23<br>Initial yield stress: 8.7 GPa |

Additional assumptions considered for the simulation are outlined as follows:

(a)   The material is considered completely homogeneous, isotropic, and defect-free.

(b) The residual stress in the deposited film is assumed to be negligible.

(c) During the simulation the material undergoes elastic deformation perfectly prior to the set in of plastic deformation.

(d) The contact between film and substrate is considered as perfectly bonded so that there is no delamination during the indentation process and the contact between film and the indenter is considered to be frictionless.

By using the 2D models, the plastic deformation behaviors of $Bi_2Se_3$ thin film on the sapphire substrate during nanoindentation can be observed. Since the system is considered to be symmetrical, only half of the test vehicle is displayed. The 2D axis symmetry finite element model with Berkovich (33,568 elements and 100,824 nodes), flat punch (18,628 elements and 55,848 nodes) and spherical (26,689 elements and 80,131 nodes) indenters are shown in Figure 1(a2, b2,c2), respectively. The PLANE 183 element type was used. The $Bi_2Se_3$ thin films, sapphire substrate and pressure indenter employ a quadrilateral element, each defined by eight nodes. The four-node quadrilateral element is used, and high mesh refinement is adopted in the vicinity of the indent because of relatively larger deformation. The contact elements CONTA 172/TARGE 169 were used in the contact area. The interactions between the indenter and specimen are modeled as a contact pair with no friction. The behavior of the indenter, which is usually made of high-hardness and high-yield stress materials such as diamond, is assumed to be rigid. A finer mesh is adopted to increase the numerical accuracy. However, this also implies that much larger computational time is needed. Thus, one needs to perform a convergence study to get a mesh that balances the accuracy and available computing resources. We did an element convergence analysis for the element model plotted in Figure 2. To improve the accuracy of the modeling and save computational time, it is important to check the mesh density and refine them accordingly. The deformation is primarily concentrated underneath the indenter and around the indented region. Thus, denser meshes around the region are used, as shown in Figure 1(a2,b2,c2). A static analysis including large deflection was performed using the commercial finite element software package ANSYS. In the constraint condition, the Y axis of the model is the axis of symmetry, and the nodes are set to be fixed in the X direction. In particular, the nodes on the bottom of the sample are set to be fixed in the Y direction. The load was simulated by applying a displacement to the indenter during the nanoindentation simulation, as indicated by the green arrows shown in Figure 1(a3,b3,c3).

The elastic/plastic properties of thin films have been successfully studied by FEM simulation and applied to various material systems [24,25]. Based on these methods, we propose the detailed procedures described below for simulating the material system. During loading, the indenter is controlled by displacement or force and the indenter is pressed to a certain maximum displacement or loading force. During unloading, the indenter tip returns to its original position at the same rate. Here, we used displacement control patterns to characterize indentation behavior. Both loading and unloading separate 5 nm at a step depth to ensure stable convergence. The *P-h* curve is obtained by the displacement of the vertical reaction force and the rigid indenter. Both the film and substrate are assumed to be homogeneous, isotropic and elastic/plastic. However, the plastic deformation does not allow full recovery, and actual recovery is related to the relaxation of elastic strain. The contact between the film and substrate is assumed to be perfect during indentation. A perfect bonding condition between the film and substrate is also assumed, and the stress is used to determine the extent of the plastic state in the materials. The equivalent yield stress is calculated by the stress tensor and the material's equivalent stress begins to yield when it reaches the yield stress. The von Mises or equivalent strain $\varepsilon_e$ is computed as:

$$\varepsilon_e = \frac{1}{1+\nu}\left(\frac{1}{2}\left[(\varepsilon_1 - \varepsilon_2)^2 + (\varepsilon_2 - \varepsilon_3)^2 + (\varepsilon_3 - \varepsilon_1)^2\right]\right)^{\frac{1}{2}} \tag{1}$$

where $\nu =$ effective Poisson's ratio.

## 3. Results and Discussion

As displayed in Figure 3, the presented FEM simulation evidently reproduces the main features of the *P-h* curve obtained from the Berkovich tip indented $Bi_2Se_3$ thin films grown on *c*-plane sapphire substrate, despite a slight offset of the indentation load value for the first pop-in event. In particular, FEM simulation clearly mimics the multiple-step feature on the loading segment of the *P-h* curve seen in real indentation measurements. Such a feature is known to closely relate to the elastic–plastic deformation and associated dislocation activities. In any case, it is apparent that good agreement between the nanoindentation measurement and FEM results can be obtained, which, in turn, allows one to extract the corresponding equivalent stresses from the nodes of the FEM *P-h* curve.

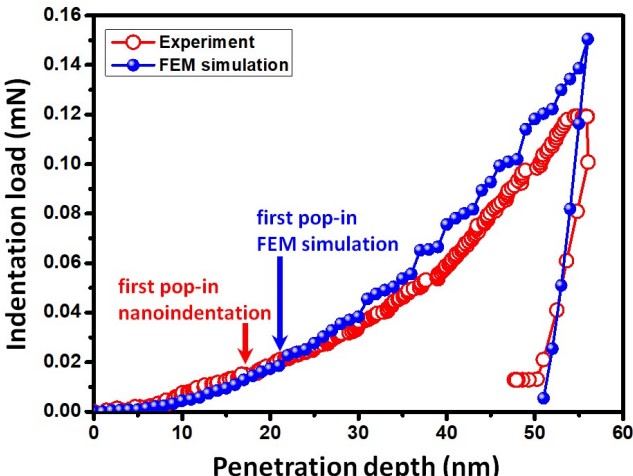

**Figure 3.** The comparsion of the experimental load-displacement curve with that obtained from FEM simulation during nanoindnetation.

Moreover, as shown in Figure 4, the behavior of penetration-depth-dependent hardness also exhibits very good agreement between the FEM simulation and the nanoindentation experimental data. It is noted that in this case the maximum indentation depth of the FEM result is 56 nm, which is only 15% of the film thickness (~360 nm) and, hence, well within the substrate-effect-free criterion suggested by Li et al. [26]. Following the Oliver-Pharr model [19], the values of *H* and *E* for the present $Bi_2Se_3$ thin film calculated from the FEM simulated *P-h* curve are about 2.1 GPa and 86.4 GPa, respectively, which are close to the experimentally recorded values of *H*~1.8 GPa and *E*~ 70 GPa.

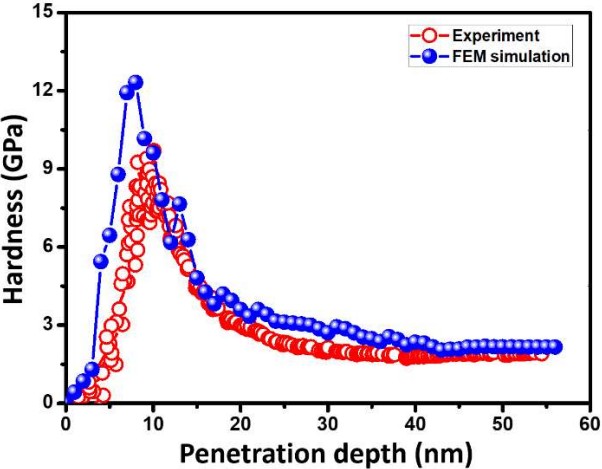

**Figure 4.** The comparsion of the experimental penetration-depth-dependent hardness and that obtained from FEM simulation during nanoindnetation.

Despite the consistency mentioned above, it is noted that there exist some subtle discrepancies between the experimental *P-h* curve and that obtained from FEM simulation. Firstly, as seen in Figure 3, the indentation load of FEM simulation starts to deviate and becomes slightly higher than the experimental value as the indentation depth is beyond 20 nm. This could be due to the fact that the regions underneath the indenter may have transformed to a more deformable structure, hence reducing the effective loading capacity of the film. On the other hand, in FEM simulation the basic mechanical property parameters of the film are assumed to remain unchanged. Moreover, as will be discussed later, the pile-up event is observed around the indentation, which may further reduce the film's loading capacity. Secondly, when the indentation depth reaches the critical depth of 20 nm, the slope of the load segment of the *P-h* curve for the FEM simulation appears to become steeper than the experimental one. This is attributed to the effect of local nanoindentation-induced dislocation nucleation behavior of film occurring beneath the Berkovich indenter tip [27]. In Figure 3, both FEM and nanoindentation curves display apparent multiple "pop-in" events, which were considered to be the signature corresponding to a clear transition from reversible elastic deformation to irreversible plastic deformation during nanoindentation and a process intimately related to dislocation activities [28,29]. In Figure 3, the first pop-in occurs at indentation depths of 17 nm and 20 nm for the experimental and FEM curves, respectively. Both are in reasonable agreement in indicating the first dislocation nucleation event. Nevertheless, as mentioned above, due to the fact that the material's parameters are assumed to remain unchanged in FEM, which may not reflect the actual film situations, the loading capacity and slope started to deviate. To gain some insight on this scenario, Figure 5 shows the contours of shear stress calculated from FEM. It gives the maximum shear stress of about 0.3 GPa, which is smaller than the nanoindentation experimental result (about 0.7 GPa). These differences can be attributed to the elastic non-linearity of the highly deformed film near the indenter tip, and to the boundary effects because of the finite size of the FEM simulated system, and can at least partially lend some support to the abovementioned arguments.

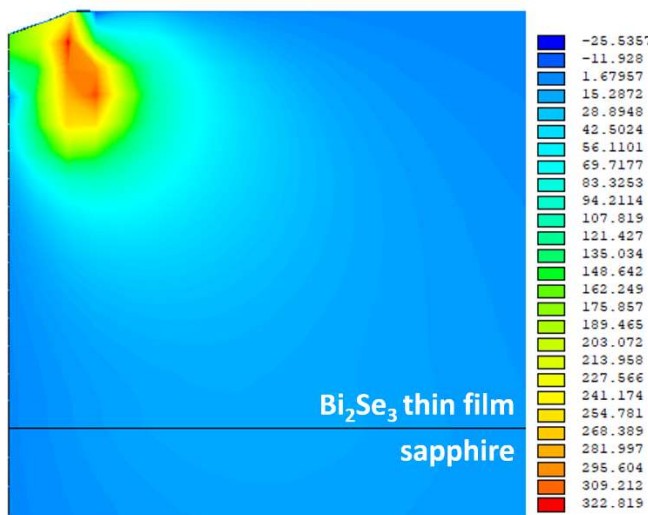

**Figure 5.** The shear stress of contours plot at first pop-in (the color bar unit: MPa).

Next, we move to discuss the nanoindentation-induced stress distribution within $Bi_2Se_3$ thin film and at the interface of film–substrate. In order to gain more comprehensive insights on this issue, the geometrical effects of various indenter tips are investigated (fixed at the same indentation depth of 56 nm). Three indenters (Berkovich, flat punch and spherical tips) are used to perform simulations of the nanoindentation process. When the diamond tip draws back, the plastic deformed region undergoes a partial elastic recovery, indicating the irreversibility of the plastic behavior during nanoindentation and indicating that the indentation-induced stress remains within the $Bi_2Se_3$ thin films. From Figure 6, it

can be found that the maximum equivalent stress is about 633 MPa in all three cases (the red part in Figure 6). Moreover, substantial nanoindentation-induced pile-up phenomena are observed with all three indenters.

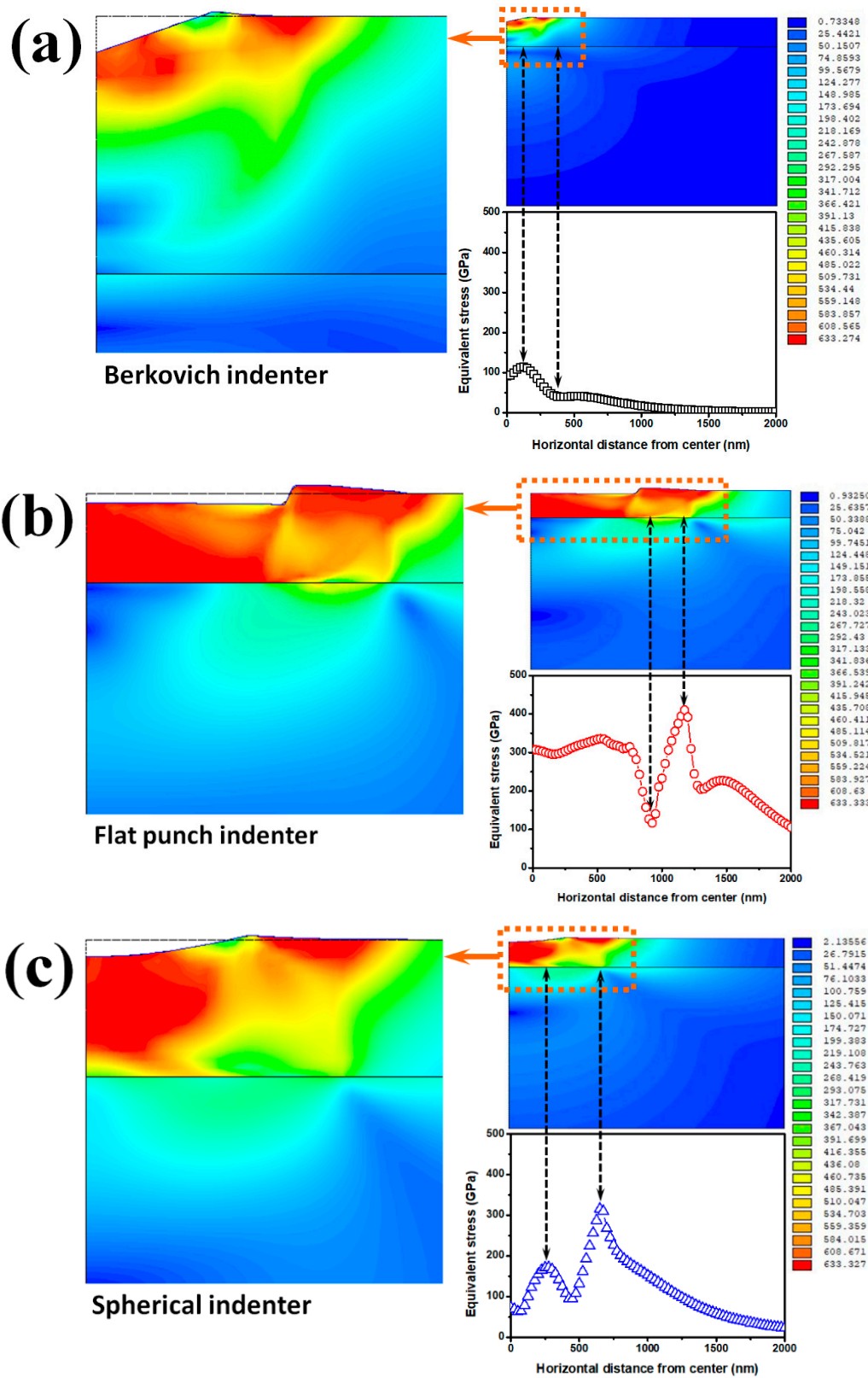

**Figure 6.** The equivalent stress distributions after unloading of (**a**) Berkovich, (**b**) flat punch and (**c**) spherical indenters (the color bar unit: MPa).

The pile-up event is often observed in systems consisting of soft film on hard substrate. For example, Tsui and Pharr [30] proposed that if the ratio of actual contact area and corner-to-corner area is greater than one, then the pile-up event can be observed, as illustrated in their experiment on an Al/glass system using a Berkovich indenter tip. A similar phenomenon was also observed in our experimental $Bi_2Se_3$ film/sapphire substrate [7]. It is interesting to note that the pile-up phenomena of modeled $Bi_2Se_3$ film/sapphire substrate can also be successfully replicated using FEM. On the other hand, it is evident that the residual stress within the $Bi_2Se_3$ film and/or the interface of $Bi_2Se_3$ film/sapphire substrate varies significantly with the geometry of the indenters (Figure 6).

To gain some quantitative perspective on the residual stress distribution caused by indenters with various geometrical symmetries, the equivalent stress curves are extracted from the interface of $Bi_2Se_3$ thin film and sapphire substrate, as displayed by color code in Figure 6. It can be seen that, for the Berkovich indenter, the residual stress is concentrated near the center of the indenter tip and surrounding the indentation (the red part in Figure 6a), and the equivalent stress is about 141 MPa exhibited at the interface between the $Bi_2Se_3$ thin film and substrate (see the corresponding plot panels shown in Figure 6a). In contrast, as shown in Figure 6b, the residual stresses are distributed quite evenly within the entire $Bi_2Se_3$ thin film when the flat punch indenter is used. Nevertheless, it is evident that the equivalent stress underneath the flat punch indenter becomes larger close to the edge of indenter (please see the zoom in the small block). Interestingly, in this case the equivalent stress of 410 MPa exhibited at the interface of the $Bi_2Se_3$ thin film/sapphire substrate is largest among the three types of indenters studied here. Finally, by comparing the results displayed in Figure 6a,c, similar deformation behaviors are observed. However, for spherical indenter tip, the zone of residual stress appears to distribute deeper and wider over the same indentation depth with an equivalent stress of 315 MPa at the interface between the $Bi_2Se_3$ thin film and substrate.

From the above discussions, it is encouraging to observe that the FEM simulations not only have evidently replicated the pop-in and pile-up events commonly observed in real indentation experiments, but also have provided means for analyzing the equivalent stress distribution within the $Bi_2Se_3$ thin film/sapphire substrate system during nanoindentation. The simulated results indicated that the deformation of $Bi_2Se_3$ thin film and the value of equivalent stress are strongly dependent on the geometry of the indenters. The present investigation indicates that the flat punch indenter tip results in the largest maximum equivalent stress (410 MPa) at the interface of the $Bi_2Se_3$ thin film/sapphire substrate system. Moreover, the simulation also confirmed that the substrate was not deformed during indentation loading and, thus, did not affect the evaluated magnitudes of hardness and Young's modulus for the $Bi_2Se_3$ thin film used in this study.

## 4. Conclusions

In summary, we report the nanoindentation responses of $Bi_2Se_3$ thin film deposited on *c*-plane sapphire substrate by combining nanoindentation measurements and FEM simulations. Results indicated that the pop-in event is displayed clearly on the loading segment of *P-h* curves in experiments and the FEM model with the Berkovich indenter tip. Moreover, the FEM simulated value of the critical indentation depth (~20 nm) is in good agreement with that of the experimentally observed pop-in depth (~17 nm). Such calculations indicate the feasibility of using FEM simulations to replicate the main features of actual nanoindentation experiments. In particular, the present study evidently demonstrated that by comparing the nanoindentation data with FEM predictions, a model for the mechanical deformation of $Bi_2Se_3$ thin films deposited on *c*-plane sapphire substrate can be reliably developed. Lastly, the comparisons performed on indenters with various geometries may also provide an efficient means to evaluate the contact-induced deformation encountered in practical device fabrication processes, wherein the shape of the contact tip may be different from case to case.

**Author Contributions:** S.-W.C., B.-S.C., P.H.L. and L.T.C.T. contributed to the experiments and FEM analyses. B.-S.C., S.-R.J., Y.-M.H., P.H.L., L.T.C.T., J.-W.L. and J.-Y.J. contributed to the discussion on materials characterizations. S.-R.J. designed the project of experiments/FEM model and drafted the manuscript. All authors have read and agreed to the published version of the manuscript.

**Funding:** The authors are thankful for financial support from the Ministry of Science and Technology, Taiwan, under Contract No. MOST 111-2221-E-214-015.

**Institutional Review Board Statement:** Not applicable.

**Informed Consent Statement:** Not applicable.

**Data Availability Statement:** Not applicable.

**Conflicts of Interest:** The authors declare no conflict of interest.

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
