# Peer review of "Finite Element Analysis of Nanoindentation Responses in Bi2Se3 Thin Films"

_coatings, doi:10.3390/coatings12101554_

Round 1

Reviewer 1 Report

In the present manuscript, the nano-indentation responses of Bi2Se3 thin film were quantitatively analyzed and simulated using the finite element method (FEM). The paper is well written and well organized. The literature is done properly. The significance of the research is highlighted very well. The results presentation is good and the graphs have been depicted in good quality with adequate discussion. The paper is interesting for readers and can be published in its present form.

Moreover, for future study, the problem can be extended to 3D model to evaluate the out-of-plane stress-strain behaviour. Also, the imperfection of bonding between film and substrate  and their crack initiation can be studied. In addition, a brittle material model can be used for sapphire material and continue the indentation till the crack initiation on the substrate and study the fracture mechanism with this material and coating Bi2Se3. 

Author Response

Dear Reviewer,

Please kindly find the attached file.

Thanks!

JIAN

Reviewer 2 Report

This is an interesting work concerning Finite element analysis of nanoindentation process. However some points needs to be addressed.

- In the end of the Introduction it should be written what is the novelty of this work in comparison to other literature works regarding modeling of nanoindentation.

-Why no friction was assumed between the indenter and the specimen? In the literature how this effect is treated? Citations may be added.

-A static analysis was selected. Why not a transient? What is approximately the duration of the real experiment of nanoindentation?

-Which equation were considered for the plasticity? Would an empirical elasto-plastic material model like Johnson-Cook would be more proper? How the stress strain curve changes after the yield point of the material?

-The term perfect contact that the authors use is the same for contact with no friction? Are any thermal effects due to friction considered?

- In figure 6 the equivalent stress should be in MPa not in GPa in the graph.

Minor errors

 Correct the phrase in line 131 "The half a 2D axis symmetry..."

In line 173 replace "by stress tensor" to become "by the stress tensor"

In line 193 replace the word excellent with very good

Author Response

(The authors gave the same response as above.)

Reviewer 3 Report

1-      FEA is deeply affected by the material properties considered during the simulation. Table 1 provides the utilized material properties. Please explain which behavior was considered for the plastic behavior of thin film and substrate. In addition, why initial yield stresses were reported? I strongly suggest to rearrange the table and give more information about the plastic behavior of the simulated materials, such as material constants or at least plastic modulus if bi-linear behavior was considered. (see https://doi.org/10.1007/s12206-021-1230-8)

2-      Authors claimed that “the contact between film and the indenter is considered to be frictionless.” Why was this assumption considered? Friction can play a significant role in the results and can be easily employed in simulations. Give a sufficient explanation or use a proper friction coefficient.

3-      Please discuss more the mesh refinement process. Was the mesh refinement performed by the automatic remeshing technique, or was the critical region manually refined? Which parameter was considered to evaluate the mesh density quality? Where is the critical region of the model?

4-      Page 3 of 11: “The behavior of the indenter, which is usually made of high-hardness and high-yield stress materials such as diamond, is assumed to be rigid.” So why the material properties of indenter were given in Table 1? Mechanical properties for a rigid material are meaningless.

5-      It is recommended to provide a magnified picture of Figs a2, b2, and c2 to show the tiny mesh under the indenter model.

6-      Give more details about the BC of the model, especially for the bottom side of the FE model.

7-      Please indicate that the given displacement in Fig. 2 belongs to which part or node. This displacement is measured along X or Y?

8-      Page 5: “During loading, the indenter is controlled by displacement or force, and the indenter is pressed to a certain maximum displacement or loading force.” Which one was used in the simulations? Displacement or force?

9-      Eq. 1 is useless and could be deleted.

10-  Fig. 3: The material has linear elastic behavior based on the assumptions and material properties. Please explain why the unloading part of FE curve is not perfectly linear.

11-  Fig. 6: please discuss why the stress contours were not obtained homogeneously, while the material was considered homogeneous

Author Response

(The authors gave the same response as above.)

Round 2

Reviewer 2 Report

The authors have adressed all my concerns.

Reviewer 3 Report

The reviewer would like to thank the authors for the applied effort in the revised manuscript.